# *In Situ* Hyperspectral Raman Imaging: A New Method to Investigate Sintering Processes of Ceramic Material at High-temperature

**Kerstin Hauke** ***, Johannes Kehren, Nadine Böhme, Sinje Zimmer and Thorsten Geisler**

Institut für Geowissenschaften, Universität Bonn, 53115 Bonn, Germany; Kehren@hs-koblenz.de (J.K.); nadine.boehme@uni-bonn.de (N.B.); zimmer@hs-koblenz.de (S.Z.); tgeisler@uni-bonn.de (T.G.)

* Correspondence: kerstinhauke@uni-bonn.de

**Featured Application:** *In situ* **high-temperature Raman imaging has a high potential to provide entirely new insights into transport and solid-solid or solid-melt reaction processes during high-temperature sintering. Therefore it help to monitor and control important technical properties of ceramic materials, including advanced and refractory ceramics, such as densification and grain growth during sintering.**

**Abstract:** In the last decades, Raman spectroscopy has become an important tool to identify and investigate minerals, gases, glasses, and organic material at room temperature. In combination with high-temperature and high-pressure devices, however, the *in situ* investigation of mineral transformation reactions and their kinetics is nowadays also possible. Here, we present a novel approach to *in situ* studies for the sintering process of silicate ceramics by hyperspectral Raman imaging. This imaging technique allows studying high-temperature solid-solid and/or solid-melt reactions spatially and temporally resolved, and opens up new avenues to study and visualize high-temperature sintering processes in multi-component systems. After describing in detail the methodology, the results of three application examples are presented and discussed. These experiments demonstrate the power of hyperspectral Raman imaging for *in situ* studies of the mechanism(s) of solid-solid or solid-melt reactions at high-temperature with a micrometer-scale resolution as well as to gain kinetic information from the temperature- and time-dependent growth and breakdown of minerals during isothermal or isochronal sintering.

**Keywords:** *in situ*; hyperspectral Raman imaging; sintering; silicate ceramics; high-temperature; $CaO$-$Al_2O_3$-$SiO_2$ system; grain growth; solid-solid reactions; spatially resolved

## 1. Introduction

In the last decades, Raman spectroscopy has become one of the most important analytical tools for a wide range of research areas in all sub-disciplines of physics, chemistry, biology, geosciences, and medicine. Initially, Raman spectroscopy was limited to scientific applications, but a low preparation effort, the ease of implementation, and developments in measurement automation have made Raman spectroscopy also interesting to users other than pure scientists. In particular, Raman spectroscopy is used for quality and process control applications [1]. In an early stage of Raman spectroscopy, the applications were limited due to generally weak Raman signals and interfering fluorescence caused by impurities of (natural) samples. However, the development of new laser technologies, filters, spectrometers, detection devices has enabled the investigation of all kinds of condensed matter. The resulting sensitivity of modern Raman spectrometer systems, combined with the high spatial resolution

of confocal optical microscopes, nowadays also facilitates to image certain properties of solid materials or liquids at the micrometer scale.

In many industries, including food processing pharmaceuticals, and biomedicine, the demand for multidimensional assessment of the sample composition has increased dramatically [2]. Hyperspectral Raman imaging enables a rapid, routinely practicable, non-destructive food quality and safety evaluation [3] and has turned into a novel clinical diagnostic tool in biomedical applications [4]. It facilitates the investigation of heterogeneous systems and thereby reveals a wealth of chemical and physical information about the chemical composition, short-range atomic structure, structural strain, crystallite orientation [5]. This information is provided with a spatial resolution down to the micrometer length scale. In earth and materials sciences, Raman imaging has been used, for example, (i) for (mineral) phase identification [6], (ii) to study internal growth structures of minerals such as zonation, (iii) to investigate the substitution of elements in solid solutions [7,8], (iv) to investigate isotope substitution and mineral replacement mechanisms [9], and (v) to *in situ* study transport and reaction phenomena during solid-water interaction at elevated temperature [10]. With the development of diamond anvil cells and heating devices, the investigation of mineral transformation reactions and their kinetics at high-temperatures and pressures became also possible. *In situ* high-temperature Raman spectroscopy has been used to study the temperature dependence of first-order phonon bands [11–14], pre-melting effects [15,16], the structure of melts [15,17–21], as well as solid-state phase transitions [22,23].

In this work, we combined the advantages of *in situ* high-temperature Raman spectroscopy with the possibility of 2-dimensional Raman imaging with a micrometer-scale resolution as a powerful tool for the *in situ* investigation of solid state sintering processes. In a recently published study, we reported the first results of sintering experiments using confocal hyperspectral Raman imaging (CHRI) to *in situ* study high-temperature, solid-state reactions in kaolin-based green bodies [24]. The most striking results of this study were (i) the formation of pseudowollastonite occurring already at temperatures as low as 840 °C, which is about 290 °C below the critical temperature for the wollastonite-pseudowollastonite transformation, and (ii) the observation that both polymorphs formed in direct contact to each other. These observations imply that pseudowollastonite can form metastably, which, for instance, raises doubts on the interpretation of the temperature conditions of pyrometamorphic rocks that is based on the presence or absence of pseudowollastonite in a rock [25]. These first results already highlighted the power of Raman spectroscopy as an analytical tool that can deliver *in situ* information about the growth and breakdown of distinct phases, including the detection of metastable or intermediate phases, without the necessity to quench the sample to room temperature (RT). Raman images can be taken during heating and cooling with a high-temperature, time, and spatial resolution, enabling the kinetics of individual reaction steps to be quantified. A single experiment, therefore, contains the information of a multitude of conventional sintering experiments, involving heating and quenching of different samples. In this work, we describe our novel approach to *in situ* study the sintering process of silicate ceramics by hyperspectral Raman imaging in little more detail. The article mainly focusses on the experimental setup, the analytical methodology, and data analysis and visualization. In addition, the power of *in situ* Raman imaging for studying high-temperature sintering reactions is briefly demonstrated by three application examples.

## 2. Materials and Method

### 2.1. Analytical Details

All Raman data presented in this work were collected with a Horiba Scientific HR 800 Vis confocal Raman spectrometer equipped with an Olympus BX41 microscope and an electron multiplying charge-coupled device (EM-CCD) detector. The Raman spectra were excited with a frequency-doubled, solid state Nd:YAG laser (532 nm) with a maximum power of 2 W, which is significantly lower at the sample surface. During its five-years lifetime, the laser lost about 20% of its power. Furthermore,

about 35% of the intensity is lost through the pathway in the instrument and additionally due to light scattering and absorption of the cell window. A 17 mW He-Ne laser was also tested, but already at temperatures of 900 °C the Raman signal was too weak and masked by black body radiation. To correct for any spectrometer shift during long-time measurements, the intense Ne line at 585.24878 (±0.00005) nm [26] that occurs at a Raman shift of 1707.06 cm$^{-1}$ in the spectra (note the typo in Stange et al. 2018 [24]) was continuously monitored by placing a Ne lamp alongside the beam path of the scattered light. A 50× long-working distance (LWD) objective with a numerical aperture (NA) of 0.5 and a working distance of 10.6 mm was used for all experiments. Because the width of a Raman band usually broadens with increasing temperature to values of up to several tenths of wavenumbers, the spectral resolution was not a major issue. Therefore, we used a grating with 600 groves/mm, so the wavenumber range from about 100 to 1750 cm$^{-1}$ could be measured in a single window. With this grating, the spectral resolution was ~3.5 cm$^{-1}$, as given by the width of the intense Ne line at 1707.06 cm$^{-1}$. The LabSpec 6.4.4.15 software was used to control the instrument, the heating, soaking, and cooling cycles, to refine the data, and eventually to create hyperspectral Raman images.

## 2.2. Experimental Series

Three types of experiments were performed in this work. The first experimental series (E1) involved heating studies with pure mineral phases that occur in silicate ceramics. These experiments were carried out to obtain high-temperature reference spectra that served as a basis for phase identification and the visualization of the Raman data from the *in situ* sintering experiments. They were performed with both single crystals (E1A) and powder pellets (E1B). Pure phase Raman spectra were taken every 10 °C from RT to 1200 °C with a total acquisition time of 100 s. The heating rate was 10 °C/min. After having recorded a spectrum at maximum temperature, the sample was cooled to RT with a cooling rate of 10 °C/min and the last spectrum was recorded. At each temperature step, depth profiles were acquired (auto-focus function) after an equilibration time of 300 s to obtain (i) an optimal signal-to-noise ratio and (ii) a measure of reaction- and temperature-related shrinking or expansion effects. One example for the latter is the decomposition reaction of calcite to lime (CaO) and CO$_2$ between about 650 and 950 °C, which could be followed by spectral depth profiling (Figure 1).

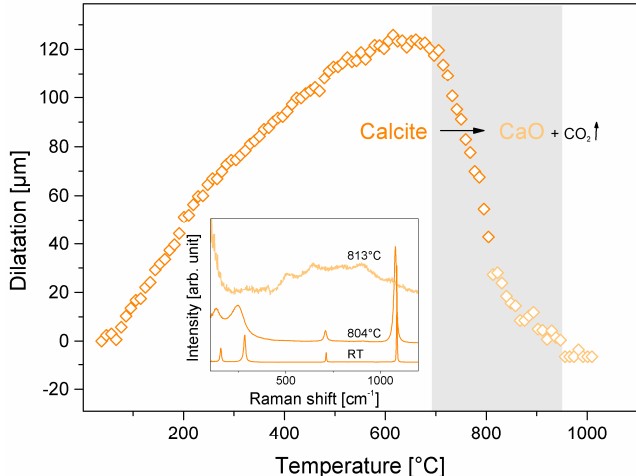

**Figure 1.** Dilatation of a pellet composed of calcite powder as a function of temperature. The dilatation was measured during the automated measurements by depth profiling. Note the dramatic shrinkage above 740 °C due to the decarbonation of calcite. The decarbonation is taking place between 700 and 950 °C, which can be followed by the shrinkage of the sample. At 804 °C, the calcite bands broaden and are shifted to lower wavenumbers (inset). Above 813 °C, calcite at the sample surface is fully decomposed to CaO (lime) that only shows a second order Raman spectrum. Note that the Raman spectra were recorded at the sample surface.

The second series of experiments (E2) involved multi-component sintering experiments and were carried out under very similar conditions as the experiments with pure reference phases. After selecting a representative area of the sample surface, a hyperspectral Raman image was first recorded at RT. The sample was then fired with a heating rate of 10 °C/min to 700 °C (corrected temperature 650 ± 5 °C, see below), where two images were subsequently recorded before heating the sample to the next temperature step. After having recorded the last high-temperature image at 1200 °C (corrected temperature 1100 ± 5 °C) the sample was cooled to RT with a cooling rate of 10 °C/min and a final image was taken.

The third series of experiments (E3) involved multi-component isothermal sintering experiments. The first image was recorded at RT to select a representative area. Then, the sample was fired with a heating rate of 10 °C/min to the target temperature of ~850 °C. Within a dwell time of 24 h, 12 Raman images (~2 h per image) were recorded. The sample was then cooled down with a cooling rate of 10 °C/min before recording the last image at RT.

*2.3. Sample Preparation*

One of the major advantages of Raman microspectroscopy is the ability to nondestructively derive spectral information with minimal sample preparation. However, the flatness of the sample surface can be crucial because light scattering at uneven surfaces may negatively affect the quality of the spectra. The samples used for the high-temperature, pure phase (E1A and E1B) and the multi-component sintering experiments (E2 and E3) were similarly prepared to minimize any systematic errors. Since high-temperature Raman spectra of single crystals (E1A) and those of their powdered counterpart (E1B) revealed differences in intensity and linewidth of fundamental Raman bands [27], the heating studies were performed with both powdered material and single crystals that were placed at the bottom of the cell close to the thermocouple. The powdered minerals used for the heating studies (E1B) were pressed into cylinders with a diameter of 3 mm and a height of less than 1 mm to minimize the temperature deviation caused by the temperature gradient in the heating stage (cf. Section 2.4). For the multi-component sintering experiments (E2 and E3) several green bodies were produced by compacting about 4.3 mg of powdered precursor material (dried at 100 °C for 24 h) into cylinders ($3 \times 2.7$ mm$^2$) at a pressure of 0.01 MPa, which resulted in a flat sample surface. The level of purity and the particle size of the precursor material were given in Table 1.

**Table 1.** Characteristics of the precursor material used for experiment E2 and E3.

| Mineral | Company | Level of Purity | Particle Size |
|---------|---------|-----------------|---------------|
| Quartz | Merck | 99.900% | 5–50 μm [1] |
| Calcite | Alfa Aesar | 99.950% | 2–20 μm [2] |
| CaO | Alfa Aesar | 99.995% | <10 μm |
| Amorphous SiO$_2$ | Alfa Aesar | 99.900% | 2–20 μm [2] |
| Kaolinite China-Clay | Carl Jäger Tonindustriebedarf GmbH | - | <2 μm (58%) 8–53 μm (8%) >58 μm (0.05%) |

[1] The initial grain size of quartz was 200 to 800 μm. The grains were milled by hand in an agate mortar to a size of approximately 5 to 50 μm. [2] The grain size was not given by the manufacturer, so the grain size was estimated from Raman images.

*2.4. Heating Stage and Temperature Calibration*

For all experiments, a TS1500 (Linkam Scientific Instruments, Surrey, UK) heating stage was mounted onto an automated x-y-z stage below the Olympus BX41 microscope objective, which facilitated to move the heating stage in all three directions with a reproducibility of ± 0.5 μm in x and y and ± 0.25 μm in the z-direction and thus to record hyperspectral Raman images at high-temperatures. The temperature of the heating device can be varied between RT and 1500 °C with temperature stability of ± 1 °C and a maximal heating rate of 200 °C/min [28]. A platinum resistor sensor, accurate

within $\pm$ 0.01 °C, measures the temperature at the bottom of the cell. The sample cup has a diameter of 7 mm and a height of 6 mm. A ceramic heating shield with a pinhole aperture of 1 mm diameter is mounted above the heating chamber to reduce the black body radiation reaching the objective and to reduce heat loss. However, due to the height and volume of the sample cylinder, a strong temperature gradient occurs within the furnace. This gradient was empirically determined by a procedure that is described in detail by Stange et al. [24], which was also used here to determine the "real" temperature at the sample surface with an error of about better than about $\pm$ 5 °C.

### 2.5. Map Programming

Depending on the exact nature of the experiment one or several hyperspectral Raman images were recorded before having fired the sample to the next temperature step. At each temperature step, a spectrum of the black body radiation was first recorded. Thereby, the acquisition parameters were identical, except that the sample was not excited by the laser. Depth profiles were acquired using the auto-focus function (depth profiling) before recording the first image at each temperature step to monitor reaction- and temperature-related shrinking or expansion effects and to obtain an optimal signal-to-noise ratio. The "tilt at limits" mode acquires depth profiles of the image center and the four corners and determines the best focus regarding the Raman signal from which the focus at each position (pixel) in the image is interpolated. Thereby an inclined sample surface can be compensated.

The quality of the Raman spectra and therefore also of the Raman images depends on a number of instrumental (e.g., laser power, acquisition time, quality of the lenses, grating, sensitivity of the detector, the size of confocal hole, the microscope objective) and sample parameters (e.g., surface roughness, color, heterogeneity, the refractive index of the sample) [29]. Raman spectra were collected during continuous x-y stage movement with a speed of 1.6 μm/s (SWIFT$^{©}$ mode). During movement, Raman intensities were recorded for 0.6 s in the wavenumber range from 100 to 1750 cm$^{-1}$. All recorded images comprised an area of 100 $\times$ 100 μm$^2$ in size. The step size was 1 μm, yielding 10,000 spectra per image. Despite the short acquisition time per spectrum, the total exposure time was still about 2 h for a single image. Hence, an image is not always an accurate snapshot, because processes may still run while recording the image. This has to be taken into account when interpreting the hyperspectral Raman images. However, reactions that are faster than image acquisition can easily be detected by line-by-line shifts in the image (Figure 2), so that the total acquisition time can be adjusted, e.g., by imaging a smaller area or by using shorter single point acquisition times.

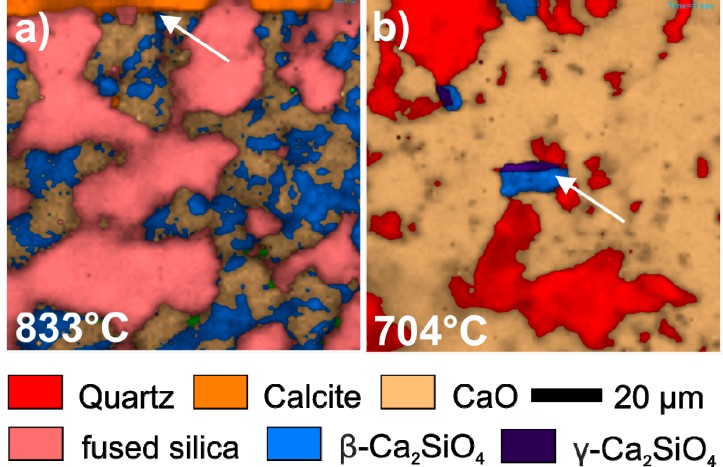

**Figure 2.** Hyperspectral Raman images of (**a**) fused silica-calcite and (**b**) quartz-lime green bodies recorded at 833 and 704 °C with a total exposure time of about 2 h, showing line-by-line image shifts. The sudden phase changes (white arrows) indicate that the respective reactions in (a), i.e., CaCO$_3$ → CaO + CO$_2$↑, and (b), i.e., γ-Ca$_2$SiO$_4$ → β-Ca$_2$SiO$_4$, were faster than the total imaging time.

In general, superior spectral quality can be obtained using a point-by-point mapping approach, as data sets can benefit from longer acquisition times. However, in this work, the hyperspectral Raman images were recorded with a continuous moving table to save imaging time and maximizing the imaged area. It is noted that even poor-quality spectra usually provide sufficient detail for phase identification. The situation will be different, however, when other band parameters than the amplitude are needed, for instance, to visualize structural or chemical variations reflected by the width and frequency of a Raman band. For Raman phase images, the quality of the reference spectra is more crucial for their quality than the signal-to-noise ratio obtained from the sample [30]. A comparison of a map recorded in the fast scanning mode (SWIFT© mode, 0.6 s per spectrum, recording time of 50 min) and in the point-by-point mode with an acquisition time of 10 s per spectrum (10 times 1 s for, recording time of 8 h 43 min) shows that the large increase in acquisition time has only a minor influence on the quality of a hyperspectral Raman image (Figure 3). A semi-quantitative estimate of the increase or decrease of a mineral fraction within the analyzed volume as a function of temperature and time is still reliable. Generally, analytical parameters (e.g., image size and EM-CCD settings) are always a compromise between (i) the required signal-to-noise ratio that depends on the spectral details to be imaged, (ii) the spatial resolution necessary to separate chemical, structural, and/or textural features, and (iii) the total imaging time that should be significantly faster than the progress of the reaction under investigation.

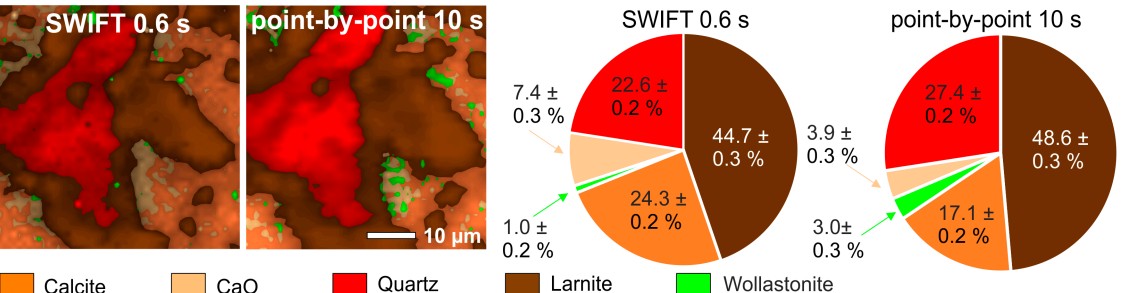

**Figure 3.** Comparison between the fast scanning mode (SWIFT© mode) with 0.6 s per spectrum (recording time 50 min) and the point-by-point method with 1 s times 10 accumulation per spectrum (recording time 8 h 43 min). Note that the acquisition time has to be faster than the mineral reactions under investigation. Grain boundaries are slightly different when using the fast scanning mode that is negligible when comparing only images recorded with the same acquisition time. A semi-quantitative statement is still possible.

## 2.6. Data reduction, Quantification, and Visualization

### 2.6.1. Data Reduction

In a first step, undesirable spectral features have to be separated from the Raman bands. This preprocessing step has to be done with care, because it is easy to introduce but not necessarily to detect artifacts in a Raman image [31]. All spectra were corrected for (i) the wavelength-depended instrumental sensitivity (white light correction), (ii) a possible spectrometer shift by using the "internal" Ne standard, (iii) cosmic spikes, and (iv) background contributions that at high-temperature are mainly composed of black body radiation and in some case of continuous fluorescence signals. As a first step, the white-light spectrum measured by the manufacturer was used to correct every spectrum of an image for the wavelength-dependent instrumental response or sensitivity. Then, the spectra were corrected for a possible spectrometer shift during long-time image acquisition using the position of the intense Ne line at 1707.06 cm$^{-1}$ of each spectrum [32]. For this, a Gaussian function was fitted to this Ne spectral line and the line position was then used to correct for any frequency shift during long-time imaging. With such an internal standard, the propagated precision of the frequency of a Raman band, obtained by least-squares fitting, was usually between $\pm$ 0.05 and $\pm$ 0.10 cm$^{-1}$ (2$\sigma$) for RT spectra. The

accuracy is thereby coupled to the precision of the Ne line wavelength determination, which is better than $\pm 10^{-5}\%$ [26].

As a next step, all spectra collected in the SWIFT$^{\copyright}$ mode, excluding the spectra taken at RT, were corrected for cosmic spikes by the spike correction function implemented in the LabSpec software. This procedure identifies all signals with a full width of half maximum (FWHM) smaller than 6 cm$^{-1}$ as cosmic signals, which are automatically removed. The analyzed spectral range was then reduced to the range between 100 and 1200 cm$^{-1}$ to simplify the following band fitting procedures. This area covers the main rocking and bending (between 100 and 700 cm$^{-1}$) and stretching vibrations (between 700 and 1200 cm$^{-1}$) of all mineral phases of interest.

For quantification and image visualization, the spectra had to be further corrected for background contributions. Two fundamental background contributions often occur within this frequency range that are related to sample fluorescence and black body radiation which is electromagnetic radiation emitted by the sample and the furnace material at temperatures above ~800 °C [33]. Fluorescence can be caused by the presence of specific fluorescing cations (e.g., $Cr^{3+}$, $Fe^{2+}$, etc.) [33] that are incorporated in the crystal structure. Fortunately, fluorescence effects are more significant at lower temperatures. The Raman spectrum of natural wollastonite, for instance, recorded at RT shows strong fluorescence signals (Figure 4), but at temperatures between 900 and 1000 °C, the Raman spectrum of wollastonite has the lowest overall background. Above 1100 °C the background is mainly influenced by the black body radiation that increases with increasing wavenumber (Figures 4 and 5). Significant portions of the black body radiation are in the region of Raman-active vibrations and therefore partly mask the Raman signals [34]. However, the black body radiation was reduced by (i) the addition of a radiation shield located between heating chamber and objective, (ii) working with a small heating system, and by (iii) decreasing the confocal pinhole, which also decreases the Raman signal.

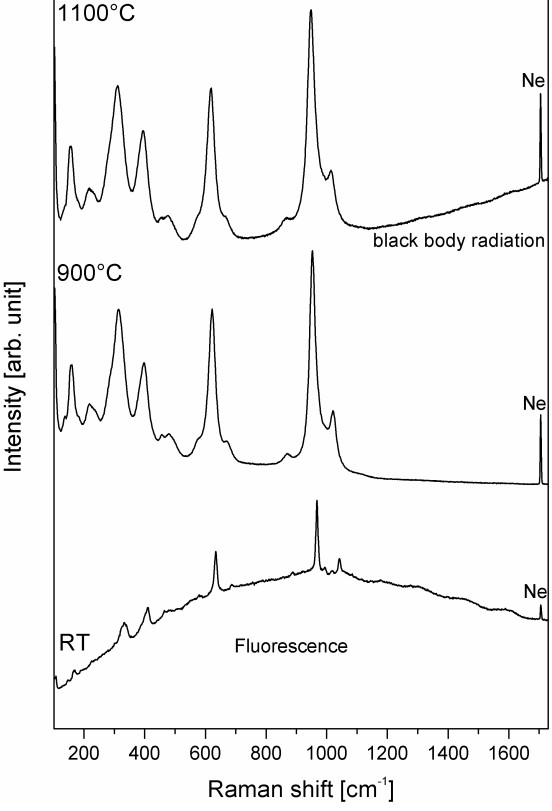

**Figure 4.** Raw spectra of wollastonite recorded at room temperature (RT), 900 °C, and 1100 °C, showing the temperature-dependent difference of the background that is dominated by fluorescence and black body radiation at low and high-temperatures, respectively.

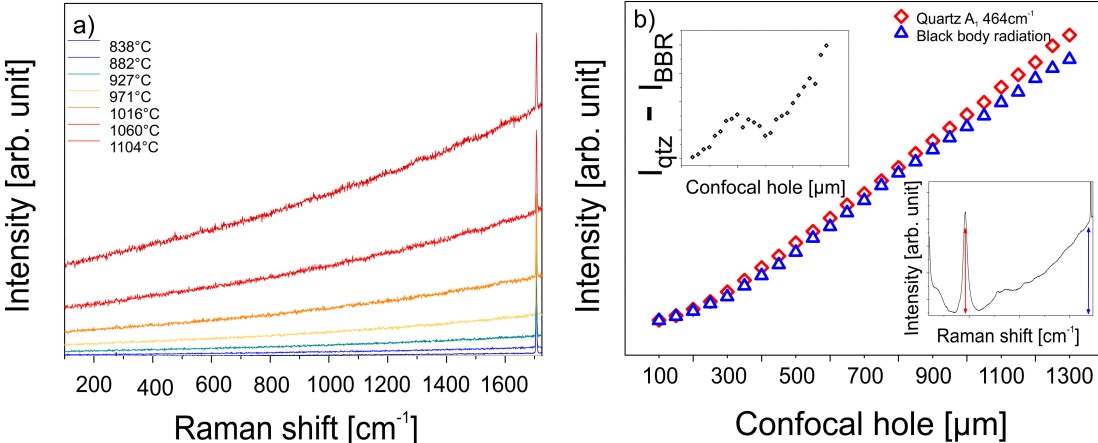

**Figure 5.** (**a**) The black body radiation as a function of temperature. (**b**) The Raman intensity of the $A_1$ quartz band near $464 \text{ cm}^{-1}$ (lower inset diagram) and the intensity of the black body radiation at $1100 \,^{\circ}\text{C}$ as a function of the confocal pinhole.

The latter effect is illustrated in Figure 5b. With increasing confocal pinhole, the intensity of the $A_1$ band of quartz near $464 \text{ cm}^{-1}$ at RT increases linearly in the range of 400 to 1300 μm. A pinhole larger than 1000 μm increases the intensity of the black body radiation to a lesser extent than the intensity of the quartz band, which leads to a better signal-to-noise ratio. Considering the intensity, closing the confocal pinhole is only a benefit if the spatial resolution of the measurement is of concern as it is for Raman imaging of multi-component sintering reactions. All measurements were thus made with a confocal pinhole of 300 μm, which was found to be a good compromise between spatial resolution, overall image recording time, and signal-to-noise ratio (cf. Section 2.7). To correct for the black body radiation, a spectrum of the black body radiation was recorded at the beginning of each temperature step. The spectrum was then subtracted from every spectrum of the image before a 2nd order polynomial baseline function was fitted to the spectrum and subtracted to further correct the spectrum for fluorescence and any background noise.

### 2.6.2. Quantification and Image Visualization

The classical least squares (CLS) fitting procedure was used to determine the phase proportion at each pixel of an image. This method is based on the assumption that a spectrum from a polyphase material is a linear mixture of the spectra from the pure phases. Within the multidimensional spectral array, the CLS fitting procedure finds a linear combination of reference spectra from the pure phases, which best fits the raw spectrum at each position (pixel). The CLS fitting procedure is explained in more detail elsewhere [24]. The resulting image is created by false-coloring each pixel of the image relative to the fraction of each component in the spectrum. In the unmixed color-coding algorithm that is used in this work, the phase with the highest intensity contribution in a Raman spectrum will be displayed in a color assigned to that particular phase. The brightness of the color displays the overall intensity of the spectrum. The analyzed hyperspectral Raman images are displayed in the "normalized scores" mode, where the proportions of the phases are normalized to 100%. Additionally, the images are smoothed, which improves the visualization of grain boundaries (compare Figure 6a,b). Such a procedure is straightforward if pure reference spectra are available. Due to the small grain size of the mineral phases and the limits of the spatial resolution, most spectra of an image are composed of contributions from multiple phases. Therefore, the temperature-dependent in-house reference spectra (E1A and E1B) were used as input reference spectra for the CLS fitting routine.

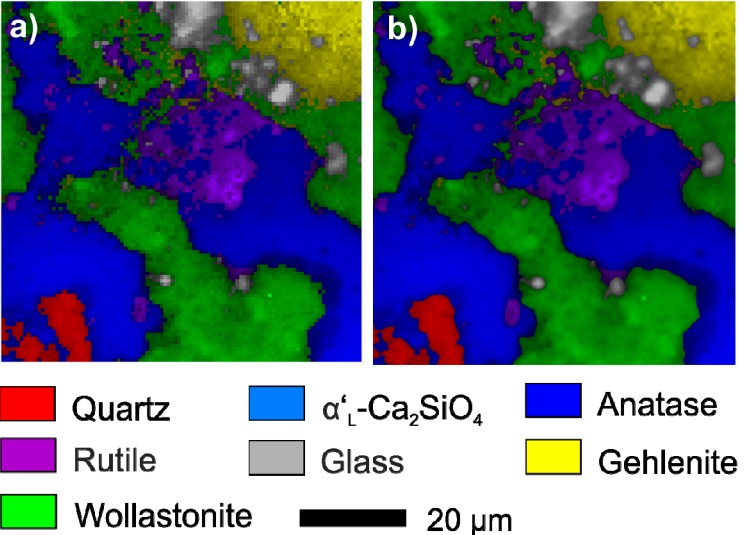

**Figure 6.** The effect of image smoothing on a hyperspectral Raman image of a sintered ceramic ($Kln_{60}Qtz_{15}Cal_{25}$, $T_{max} = 1097\ °C$). (**a**) unsmoothed and (**b**) smoothed image. Note that in particular grain boundaries are better displayed in the smoothed image.

### 2.6.3. Factors Affecting Phase Quantification

Phase quantification is influenced by four main factors that have to be taken into account when interpreting hyperspectral Raman images, namely (i) crystal orientation, (ii) grain size, (iii) temperature dependence of band position and linewidth, and (iv) the Raman scattering cross-sections of the mineral phases.

For anisotropic crystals, the intensity of a Raman band depends on the angle of incidence of the laser beam in relation to crystal lattice orientation, bond polarizabilities, and the state of polarization of the incident laser beam and therefore relative band intensities of a mineral phase change with crystal orientation, whereas the band shape and position is independent of crystal orientation. Room-temperature, unpolarized Raman spectra of fibrous Kopparberg wollastonite collected with the crystal fiber axis parallel and vertical to the electric field vector of the incident light are very similar, i.e., only the weak Raman bands are strongly polarized [35]. In contrast, due to the symmetry of the crystal structure, Raman spectra of rutile oriented with the crystal c-axis parallel and perpendicular to the polarization of the laser beam vary strongly [36]. It is clear that any differences in relative Raman band intensities due to the orientation of the crystal affects the CLS fitting procedure. However, most of the mineral phases occurring during the experiments presented here exhibit limited orientation effects that do not significantly affect the fitting result. Nevertheless, for minerals with relative large orientation effects like gehlenite, reference spectra were recorded in a number of different orientations and implemented in the CLS fitting procedure. A positive side effect is that the orientation effect can be used to systematically study grain rotation during firing, which is particularly important for our understanding of the dynamics of crystal growth processes in ceramics with potential for certain device applications [37]. With Raman spectroscopy relative crystal orientations can be determined rapidly and accurately without the necessity of sample preparation [37].

Raman spectra of single crystals compared to those of powders reveal differences in linewidth and intensity of the fundamental Raman bands (Figure 7). In addition, the spectra of some mineral powders show additional bands which are likely due to surface scattering [27]. The former can be due to local heating and/or quantum confinement [38], whereby the latter becomes significant only in nanocrystalline materials and here mainly effects the lattice modes. The dependence of linewidth on grain size was already studied, for instance, by several authors for $TiO_2$ nanocrystals [39–41] and nanometer-sized diamonds [38,42,43]. Zhao et al. [44] studied the influence of the size of diamond particles, ranging in size from 0.25 to 10 $\mu$m, and the beam power, varied between 8 to 100 mW, on

the fundamental diamond Raman band near 1332 cm$^{-1}$. They explained an observed systematic downshift of the Raman band frequency as reflecting both the decrease of particle size and a local temperature increase of about 500 °C, which depends on the thermal conductivity path available in the specific configuration [44]. Therefore, the reference spectra used for the CLS fitting procedure were mainly obtained from pellets made from powdered samples with grain sizes between 5 and 30 μm (E1B) rather than from single crystals.

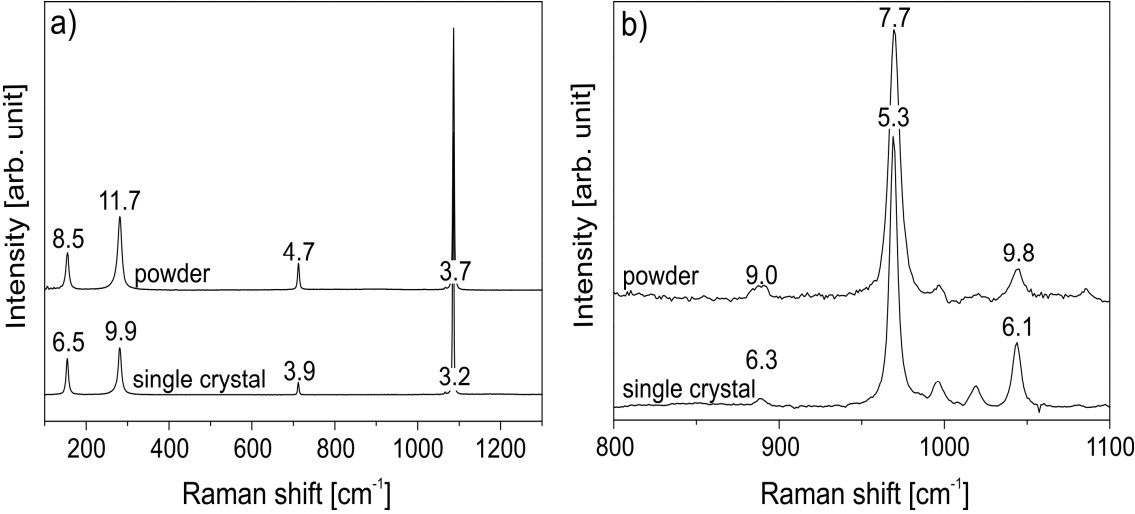

**Figure 7.** Comparison of Raman spectra from single crystals (E1A) and powders (E1B) of (**a**) calcite and (**b**) wollastonite. Numbers give the full width at half maximum (FWHM) of the respective band. Note that the Raman bands measured on powders are characterized by larger FWHM values.

The temperature-dependent mode behavior of most of the relevant phases in silicate ceramics has already been studied by Raman spectroscopy, such as albite [22], wollastonite [35], gehlenite [15], diopside [45], pseudowollastonite [15,45], and quartz [46]. However, the high-temperature Raman spectra reported in these studies were recorded with different instruments and under different analytical and experimental conditions (e.g., larger temperature steps) and are also not precise enough. We, therefore, collected in-house, high-precision, high-temperature reference Raman spectra of the relevant ceramic phases. These were used for the estimation of relative Raman cross sections that are needed (i) to determine the relative phase proportions from a mixed Raman spectrum, and (ii) to identify individual high-temperature phases in convoluted spectra.

The transverse Raman scattering cross-section depends on many factors like pressure [47], temperature [48], and the crystal orientation with respect to the polarization direction of the incoming laser light [49]. The correct calculation of Raman intensities in the context of the polarization theory is difficult and involves an enormous calculation effort. Therefore, theoretical calculations of Raman intensities were less commonly performed than calculations of infrared or nuclear magnetic resonance spectra [50]. It is common practice to normalize Raman spectra to the maximum intensity [18,51–53] or to divide the intensity of the Raman spectra by the power of the laser and the acquisition time [52,54] to compare different Raman spectra. Therefore, in our first attempt [24], we normalized the reference spectra between 0 and 1 before using them for CLS fitting (Figure 8a). However, this procedure neglects the Raman cross sections of the mineral phases. In the present work, we, therefore, recorded the reference spectra at the same measurement conditions (i.e., laser wavelength and power, acquisition time, internal Ne lines, etc.) and used the processed reference spectra without normalizing the intensity for the CLS fitting procedure (Figure 8b). However, this approach does not consider all factors that affect the Raman intensities. The main source of errors, aside of instrument calibration errors, is related to the unpredictable loss of light at uneven surfaces by surface scattering. However, due to the equal procedure during data treatment, it is still possible to semi-quantify changes of mineral fractions and

to monitor grain boundary movements during the experiment. From multiple Raman images of the same area of a fired CaO-SiO₂-Al₂O₃ ceramic ($T_{max}$ = 1097 °C), recorded using the same analytical parameters as for the *in situ* experiments, we obtained empirical errors in order of $\pm 20$ and $\pm 1\%$ at a volume fraction of 0.3 and 75.00%, respectively, that scale with the square root of the average mineral fraction within the analyzed area (volume), as expected for errors relating to the Poisson statistics of counting rates.

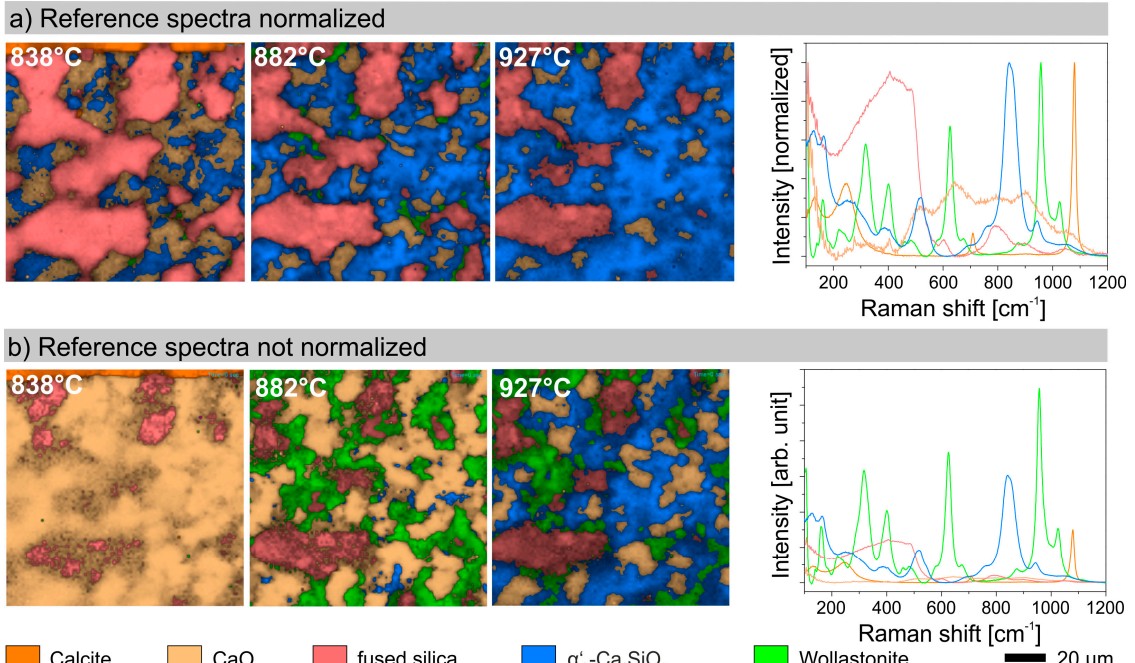

**Figure 8.** A series of three hyperspectral Raman images taken between 838 °C and 927 °C which are generated with the classical least-squares (CLS) fitting method using in-house reference spectra that are (**a**) normalized and (**b**) not normalized before loading. The reference spectra in (b) are recorded at the same measurement setup and thereby the Raman cross-sections of the individual mineral phases are considered. Note that the normalization of the reference spectra causes mineral phases with low Raman scattering cross-sections such as CaO to be underrepresented in the image.

## 2.7. Lateral and Axial Resolution

The spatial resolution is an essential factor for interpreting hyperspectral Raman images. Unfortunately, only a few studies have dealt with this topic until now [5,55–60]. The spatial resolution of Raman microspectroscopy is governed mainly by the diffraction limit of light, and therefore dependents on the laser wavelength ($\lambda$) as well as on the numerical aperture (*NA*) of the objective [5]. In a first approximation, the theoretical lateral ($d_l$) and axial ($d_a$) resolution is linearly dependent on the wavelength of the incident laser and inversely proportional to the numerical aperture of the objective. It follows that a high spatial resolution can be achieved with lasers at shorter wavelengths and a high-magnification optics.

With the 50× long-working distance objective (*NA* = 0.5) used in this study, the diffraction limited theoretical lateral and axial resolution at the sample surface is 1.3 and 8.5 µm, as given by $d_l \approx 1.22 \, \lambda/NA$ and $d_a \approx 4\lambda/NA^2$, respectively, where $\lambda$ is the excitation wavelength [61]. In practice, the real resolution is certainly worse than the theoretically calculated values due to (i) an imperfect optics, (ii) light scattering at the surface and interfaces of the sample, and (iii) light refraction at the top window of the heating stage. The axial resolution can principally be improved by using a confocal aperture, reducing the volume from which the scattered light is collected by blocking radiation generated from the surrounding volume [55]. A convenient side effect is the reduction of the black body radiation, which is emitted by the sample at elevated temperatures by reducing the confocal

aperture. A drawback of narrowing the confocal hole is that the overall intensity becomes lower (Figure 5).

To interpret hyperspectral Raman images and especially to interpret the spectrum arising from a specific point in the map (for example grain boundaries), it is essential to know how much of the observed Raman signal originates within the focal volume and how the response tails off with depth [57]. Two factors are primarily important for the depth resolution, the volume of the laser focus, and how Raman photons generated within this volume are relayed back into the spectrometer via the confocal aperture [5]. The axial resolution obtainable in confocal microscopy can be calculated by the following equation [62]:

$$d_a = \frac{2.2n\lambda}{2\pi NA^2} \tag{1}$$

where $n$ is the refractive index of the immersion medium. This equation is only valid with the optical focus at the surface of the sample.

Based on the theoretic consideration, Everall [5] showed that the true laser focus is always deeper within a transparent sample than the so-called nominal depth ($\Delta$), i.e., the distance below surface given by the $z$ drive of the stage. The axial laser focus also broadens upon moving deeper into the sample. The depth resolution gets worse linearly with the nominal depth (distance below the surface) [55] and can be estimated from:

$$d_a = \Delta \left[ \left[ \frac{NA^2 (n^2 - 1)}{(1 - NA^2)} + n^2 \right]^{\frac{1}{2}} - n \right] \tag{2}$$

Apart from the point of focus also the depth of focus increases with increasing nominal depth, so it becomes impossible to obtain "pure" spectra at interfaces [61].

To study the influence of the confocal hole on the depth resolution, 13 hyperspectral Raman images of a fired $CaO$-$SiO_2$-$Al_2O_3$ ceramic ($T_{max}$ 1097 °C) were recorded at RT at the same location with a different confocal pinhole (100 to 1300 μm). With decreasing pinhole, the depth resolution increases, but at the price of intensity loss (cf. Figures 5b and 9). Small rutile grains (purple) are clearly distinguishable within the Raman images recorded with a confocal hole of up to 400 μm, whereas they appear to have "grown" together in images recorded with a larger confocal aperture (Figure 9). Furthermore, due to the overall lower intensity of the images recorded with a confocal hole smaller than 300 μm most of the grain boundaries are blurry. As already shown in Figure 5b, the intensity decrease with decreasing confocal hole is linear over a large range with a slight kink at a confocal hole of 300 μm. For our Raman system and the chosen instrumental settings, a confocal hole of 300 μm appears to be the best compromise between sufficient intensity and spatial resolution.

It is important to note that the axial resolution at the surface also depends on the absorption properties of the material investigated. To study this dependence, a series of hyperspectral Raman images of a fired ceramic ($Kln_{60}Qtz_{15}Cal_{25}$, $T_{max}$ 1097 ± 5 °C) were recorded at different nominal depth ($z$ drive) to investigate the effect of the laser focus on the quality and details of the image (inset of Figure 10). As expected, the intensity of the Raman bands of the individual phases decreases with increasing nominal depth (Figure 10). A Raman intensity loss of 20% to 40% is observed within the first ten micrometers, whereas within the first 20 μm, which is a typical grain diameter, already about half of the intensity of the Raman bands is lost. At a depth of about 50 μm, Raman signals from any of the analyzed phases are hardly detectable anymore due to strong absorption of the incident and scattered light.

This is also reflected by the observation that the Raman images recorded at different nominal depths from the surface ($z = 0$ μm) to a depth of $z = -15$ μm are very similar (Figure 11). As a matter of fact, the statistical variations of the average mineral fractions, obtained from all 16 images, are in the same order of magnitude of those that were estimated from multiple images that were recorded from the same area (volume) with identical focus (cf. Section 2.6.3). It follows that slight variations of the focus among different images (e.g., due to failing autofocus, shrinking or expansion of the sample

during image acquisition) do not significantly affect the recorded mineral fractions and morphological details (e.g., the location of grain boundaries).

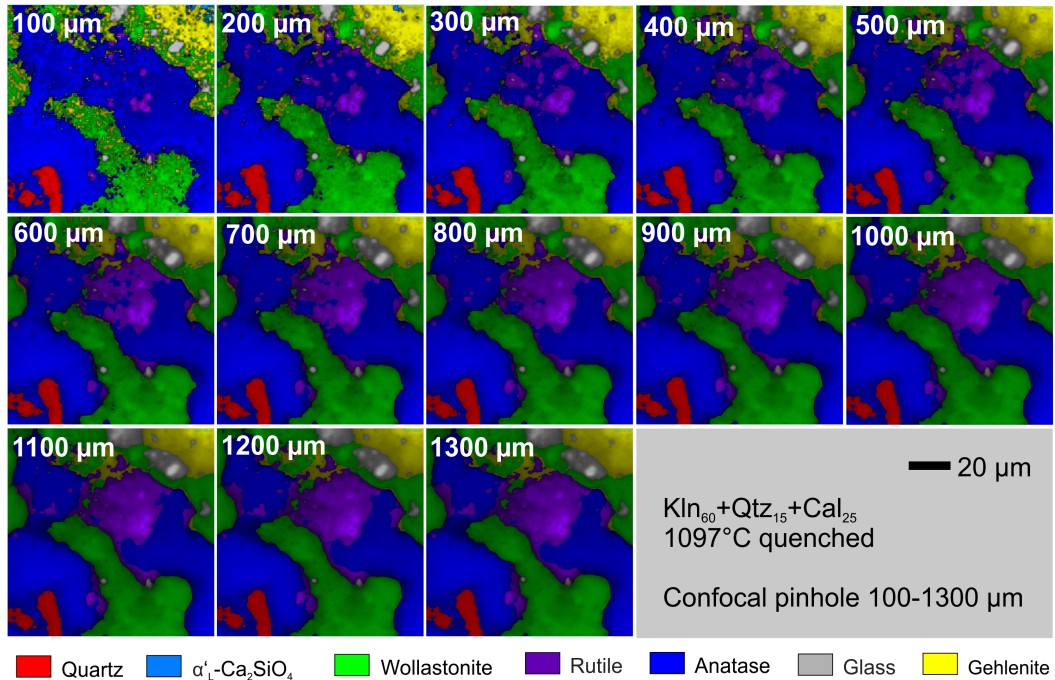

**Figure 9.** RT hyperspectral Raman images of a fired CaO-SiO$_2$-Al$_2$O$_3$ ceramic recorded with a different confocal pinhole. Note that images acquired with a pinhole smaller than 300 μm show the worst quality due to significantly lower band intensities, whereas with larger confocal pinholes morphological details get lost. For instance, the small rutile grains (purple) that are clearly distinguishable in those images recorded with a pinhole smaller than 400 μm are increasingly "grown" together and appear to be one single grain in the images recorded with a confocal pinhole larger than about 800 μm.

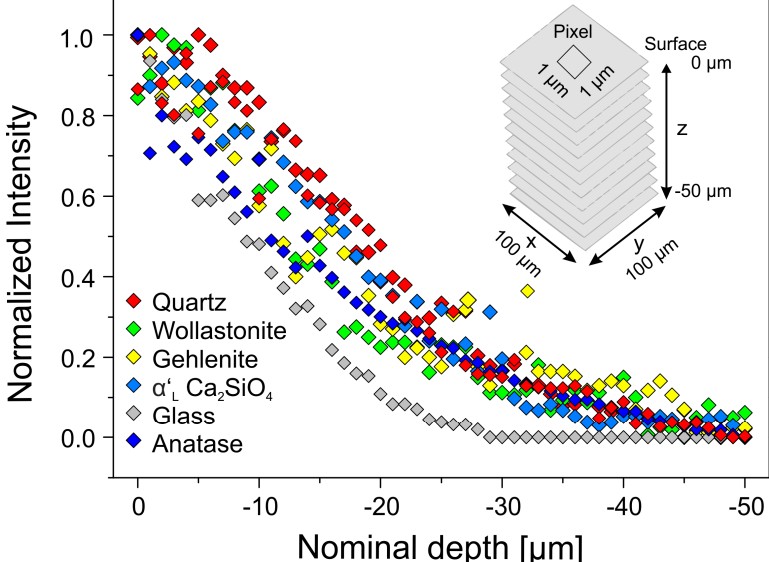

**Figure 10.** Normalized intensities of the main Raman band of the respective phase at six x-y positions as a function of nominal depth (*z* = 0, focus at the sample surface). The data was extracted from 50 hyperspectral Raman images of a fired ceramic that were recorded at RT at a focal depth between 0 and −50 μm (*z*-axis) with a step size of 1 μm (inset diagram) and a confocal hole of 300 μm.

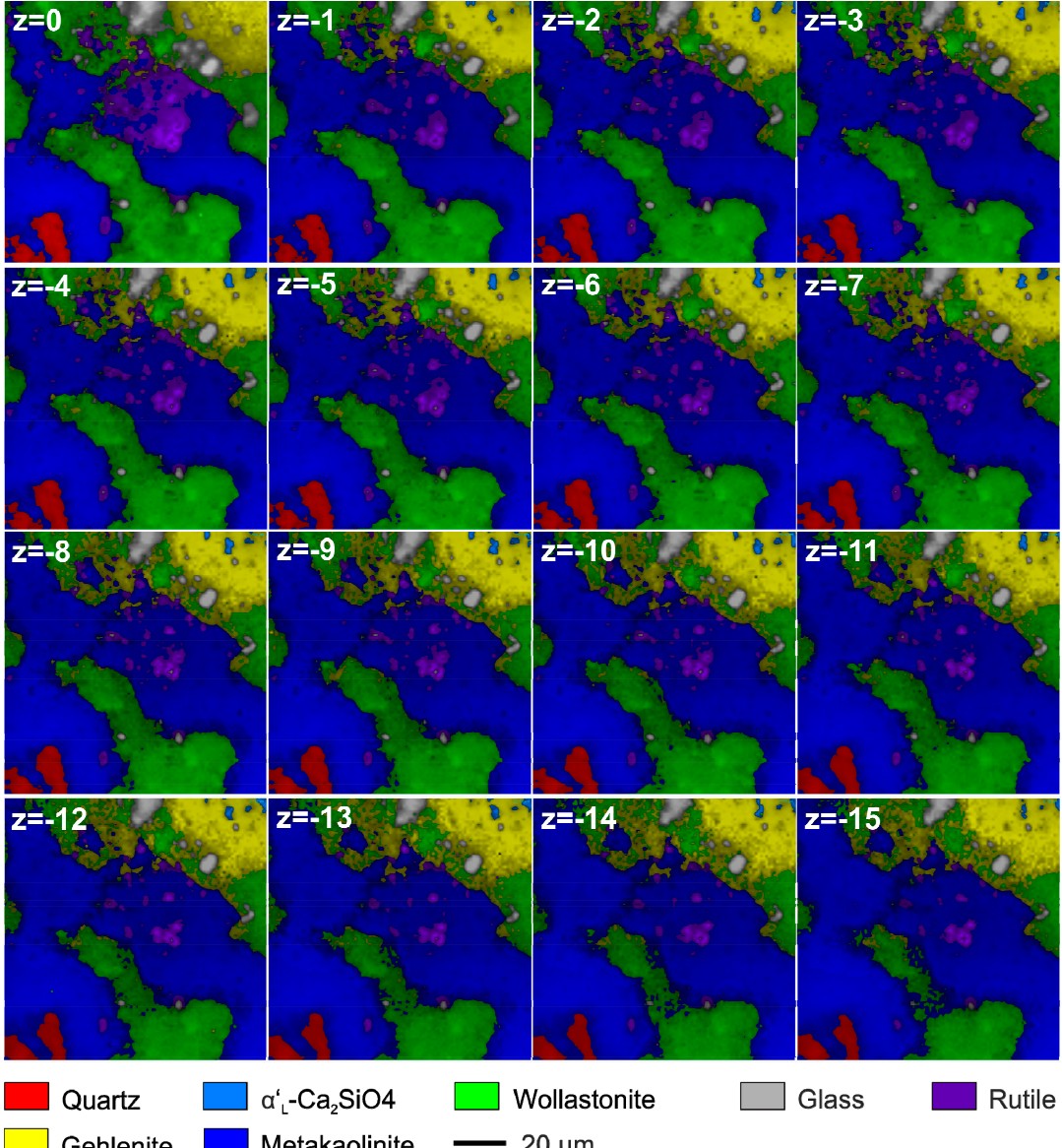

**Figure 11.** A series of 16 hyperspectral Raman images of a fired ceramic that were recorded with a nominal depth $z = 0$ (focus at the surface) and $z = -15$ μm (focus 15 μm below the surface). Note that despite the significant offset of the focus all images exhibit the same relative phase proportions and main morphological details.

## 3. Applications

### 3.1. Mineral Reactions During Firing and Cooling

The occurrence or non-occurrence of certain mineral phases and their relative proportions are often used, for instance, to (i) determine the firing temperature of ancient ceramics, (ii) to divide ancient ceramics into groups for provenance studies [63], (iii) to estimate the formation temperature of natural rocks [25], or (iv) to determine the firing temperature of brown coal ashes burnt in a power plant [64]. The difficulty here is that it is not always obvious whether a mineral formed during firing or during cooling. Tschegg and coworkers [63], for example, could not solve the question whether large sparitic carbonate inclusions, observed in Cypriot Bronze-age plain white ceramics, represent incompletely decomposed or secondary calcites that were formed by a recarbonation process after firing. Such questions can principally be answered by *in situ* experiments [24].

To study the decarbonation and possible recarbonation process, i.e., the influence of cooling on the final product, a sample composed of quartz and calcite (mass ratio 1:1.03) was fired with a heating rate of 10 °C/min and 50 °C temperature steps from 660 ± 5 °C to 1106 ± 5 °C. At each temperature step two hyperspectral Raman images ($100 \times 100$ µm$^2$, 1 µm step size, SWIFT 1.6 µm/s) were successively recorded within a dwell time of four hours. After recording the last hyperspectral Raman image at the maximum temperature of 1106 °C, the sample was cooled to RT with a cooling rate of 10 °C/min.

At 839 ± 5 °C calcite gradually decomposes to CaO (images not shown here, but see Figure 1). Calcium oxide has a rock-salt structure and thus no first-order Raman spectrum [65], but exhibits second-order Raman features. Second-order Raman features are rarely used for Raman imaging so far [60]. However, CaO could be identified by Raman features near 540, 670, 740, and 900 cm$^{-1}$ and house-intern reference spectra of CaO were successfully used for data processing.

With increasing temperature $\alpha'_L$-Ca$_2$SiO$_4$ formed from a reaction between lime and quartz. During cooling $\alpha'_L$-Ca$_2$SiO$_4$ transformed to β-Ca$_2$SiO$_4$ that remained metastable at RT. Importantly, during cooling calcite recrystallized from a reaction of lime with atmospheric CO$_2$ which is released by the calcite decomposition during heating and partially trapped in the heating stage. This is noticed when comparing the *in situ* Raman image recorded at 1106 ± 5 °C with that recorded at RT, clearly showing that the calcite within the imaged area has fully decomposed to CaO and CO$_2$ at 1106 °C (Figure 12). However, without this *in situ* information, one could come to the conclusion that the calcite grains observed in the burnt ceramic represent remnant precursor calcite grains. It follows that *ex situ* quench experiments are not necessarily sufficient to fully investigate mineral reactions at high-temperatures during firing and subsequent cooling. However, it should be noted that only the sample surface is analyzed by Raman imaging. Therefore, based on the Raman data alone, it cannot be ruled out that only CaO from the first few micrometers recrystallizes to calcite because CO$_2$ cannot reach the interior of the ceramic body.

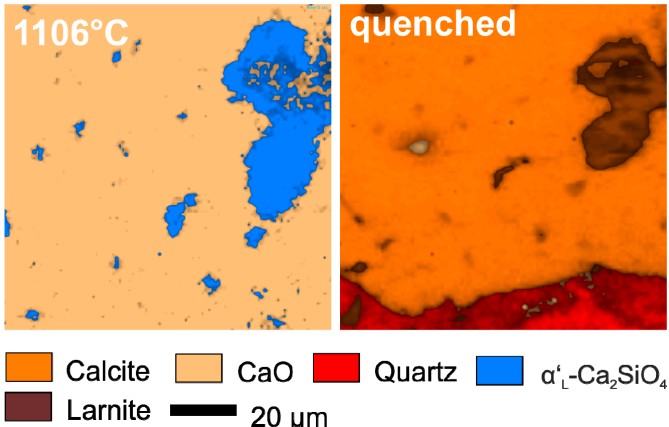

**Figure 12.** Raman images of a sample composed of quartz and calcite fired to the last temperature step of 1106 °C (**a**) and quenched to RT (**b**). At 1106 °C calcite is decomposed to CaO and CO$_2$ and quartz completely reacted to $\alpha'_L$-Ca$_2$SiO$_4$ (a). After cooling with a cooling rate of 10°C/min to RT lime has reacted with atmospheric CO$_2$ to calcite (b). During cooling, $\alpha'_L$-Ca$_2$SiO$_4$ has transformed into β-Ca$_2$SiO$_4$ (larnite). Note the slight offset and grain boundary shifts, which is due to differential shrinkage of the sample during cooling.

### 3.2. Isothermal Mineral Reactions and Grain Growth

Understanding the kinetics of sintering reactions is necessary to systematically control the manufacturing process of a ceramic, but also to understand metamorphic rock-forming processes in nature. Even if a mineral transformation is thermodynamically favored, it is not guaranteed that the transformation will occur at any measurable rate. The reaction rate strongly depends on the mechanisms involved in the reaction process and can change with time and temperature. Grain

boundaries move and rearrange so as to increase the average size and decrease the grain boundary area per unit volume [66]. Here, isothermal *in situ* experiments were performed to study the effect of time on the formation of calcium silicates and grain growth during firing. One major advantage over conventional quench experiments is that the same sample position/material is continuously investigated without the need for quenching a new sample at each time step.

For the experiment, a mixture of quartz and CaO (mass ratio 1:1) was fired with a heating rate of 10 °C to 848 ± 5 °C and 12 hyperspectral Raman images (100 × 100 μm, 1 μm step size, SWIFT 1.6 μm/s) were recorded step by step within a dwell time of about 24 h. After the last hyperspectral Raman image had been taken, the sample was cooled to RT with a cooling rate of 10 °C/min.

During the first 10 h, the sample has significantly shrunk, which is reflected by a shift of the imaged area (Figure 13). Lin and co-authors also observed increased shrinkage within the first 10 h of isothermal sintering of $CaSiO_3$ at 1100 ± 5 °C and explained this phenomenon by incomplete densification and increased porosity of the samples during the first few hours [67]. In order to semi-quantify the apparent mineral content and grain size distribution over time, an area of 80 × 80 $μm^2$ was selected, which is more or less present in each image (white dotted squares Figure 13). For quantification, the marked area was used. Note that due to the three-dimensional shrinking of the sample, a complete match cannot be achieved. During the recording of the first hyperspectral Raman image at 848 ± 5 °C calcite was still decomposing to CaO and $CO_2$ (Figure 13). Note that in this case, the decomposition reaction was significantly faster than the image recording time (cf. Figure 2), reflected by the observation that calcite is still visible within the first rows of the image, but is not detected anymore afterwards. At the same time, $α'_L$-$Ca_2SiO_4$ formed from a solid-state reaction between lime and quartz, which transformed to larnite during cooling (cf. RT image in Figure 13). The fraction of $α'_L$-$Ca_2SiO_4$ increased linearly with time, while simultaneously the fractions of the CaO and quartz decreased also linearly (Figure 14).

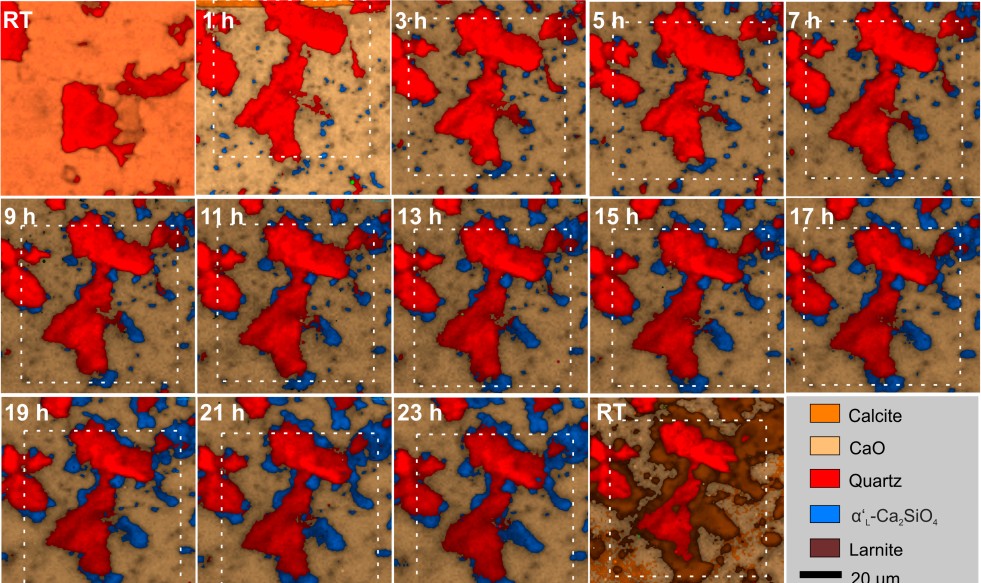

**Figure 13.** Hyperspectral Raman images of a green body composed of a mixture of quartz and CaO (mass ratio 1:1). The sample was fired with a heating rate of 10 °C to 848 ± 5 °C and 12 hyperspectral Raman images (100 × 100 μm, 1 μm step size, SWIFT 1.6 μm/s) were subsequently recorded within a dwell time of about 24 h to follow the isothermal reaction kinetics. Calcite still decomposed during recording the first Raman image. Simultaneously, $α'_L$-$Ca_2SiO_4$ formed from CaO and quartz. With increasing sintering time, the content of $α'_L$-$Ca_2SiO_4$ increased. During cooling, $α'_L$-$Ca_2SiO_4$ transformed to larnite and remained metastably at RT. White rectangles mark the areas used for semi-quantitative determination of the phase composition (see text).

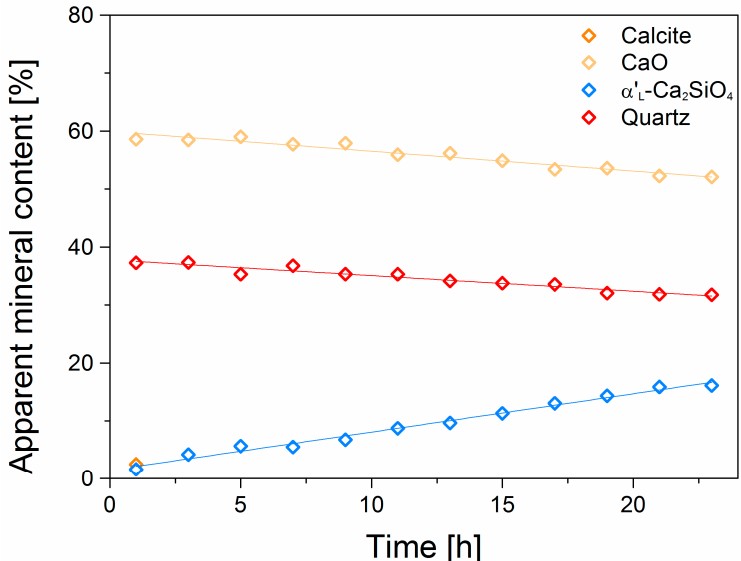

**Figure 14.** The evolution of the mineral fractions as a function of time during isothermal sintering of a quartz-calcite (mass ratio 1:1) green body at 848 °C obtained by the binary images of the single phases. The colored lines represent a linear fit to the data over the whole time interval.

The linear decrease of the reactant fractions with time suggests zero-order kinetics, which, however, is always a result or a kind of artifact of the conditions under which the reaction is carried out. For this reason, reactions that follow zero-order kinetics are often referred to as pseudo-zero-order reactions. A pseudo-zero-order behavior principally suggests that either only a small fraction of the reactants was able to react, which is continually replenished, or that the concentration of one reactant was much larger than those of the others. In our case, the latter explanation is more likely since the investigated area is dominated by lime that occurs with a volume fraction of about 60% in our experiment. Note that such a zero-order process cannot continue after one reactant has been exhausted or is protected against reactants by a solid reaction rim. If this point is reached, the reaction will change to another rate law such, for instance, the Avrami law [68], commonly used to describe the isothermal kinetics of phase and microstructure transformations, instead of falling directly to zero. In our experiments, this point was not reached within 24 h of sintering.

In addition, Raman imaging also enables monitoring of single grain growth during ceramic firing. Understanding grain growth is important to nearly every engineered material as porosity and grain size affect the strength of polycrystalline ceramics in a similar manner [69]. The strength of ceramics decreases with an increase in porosity and grain size [67] and therefore a stable small grain size is desirable for materials that rely on strength, toughness, or formability [66]. For grain size analyses, an area was selected, where newly formed $\alpha'_L$-Ca$_2$SiO$_4$ is present. By using a color threshold, these image areas were converted into binary images with the aid of the graphics program ImageJ [70], which were then used to quantify the evolution of grain size of $\alpha'_L$ Ca$_2$SiO$_4$ as a function of time (data not shown here) by applying the "Analyze Particles" tool from ImageJ. With increasing sintering time, the grains of $\alpha'_L$-Ca$_2$SiO$_4$ have grown and small grains have partly coalesced, whereby the average grain size (Figure 15, black squares) increased linearly over time, also indicating a pseudo-zero-order behavior. This supports the interpretation that the pseudo-zero-order behavior is due to the fact that the concentration of one reactant (CaO) was much greater than that of the others (quartz). In addition, the size of most single grains (e.g., grain 2, 3, 5, 6 in Figure 15) have increased linearly with time, whereas some grains (e.g., grains 1 and 4 in Figure 15) did not significantly grow at all. These grains must have been separated from a continuous flux of SiO$_2$ or CaO for some reasons and therefore could not grow further. The growth of grain 5, however, was not limited, since several small quartz grains were surrounded by a CaO matrix and thus the contact surfaces between CaO and SiO$_2$ were very large.

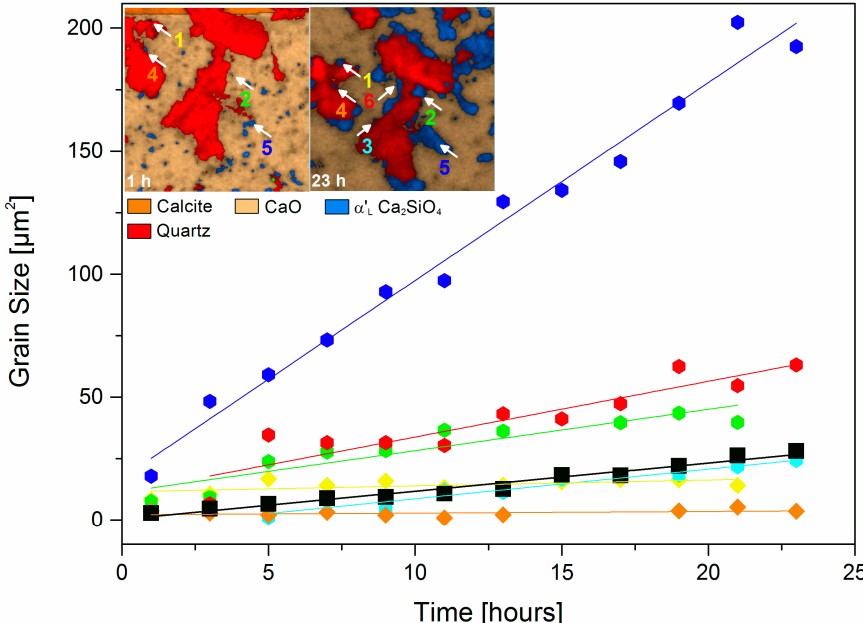

**Figure 15.** The growth of individual grains as a function of time. The average grain size (black squares) increases linearly with increasing sintering time which corresponds to a pseudo-zero-order behavior. In addition, the size of most single grains (colored hexahedron, grain 2, 3, 5, 6) increases linearly with time. The lines represent linear least-squares fits to the data. Image size in the inset is $100 \times 100 \ \mu m^2$.

This example demonstrates that the growth of distinct grains can be investigated over time without the need to use a new sample for each time step, which is a major advantage over conventional quench experiments. This enables studying the mechanism(s) of nucleation and growth, the quantification of grain growth rates, the identification of rate-limiting reaction steps, and potentially the determination of activation energies from isochronal heating experiments.

### 3.3. In Situ Observations of the Migration of Solid-Solid Reaction Fronts

Reaction rims and corona textures are common non-equilibrium features in many metamorphic rocks [71,72] and ancient ceramics [63]. As the reactant and product phases have different compositions, interlayer growth requires chemical mass-transfer which, in the absence of melts and fluids, can occur by diffusion only. At the same time, localized reactions must proceed at the 'reaction interfaces', which divide the growing layers on either side [72]. Both processes may be rate limiting and the coupling between the two processes determines the overall reaction kinetics [72]. Two approaches are possible to identify the rate-limiting step and to identify the direction of the migration front: (i) the rim growth approach and (ii) experiments with isotopically doped reactants [71,73,74].

To study the solid-state reaction between quartz and lime at the grain scale at high-temperatures, a mixture of quartz and CaO (mass ratio 0.93:1) was fired with a heating rate of 10°C/min stepwise (50 °C steps) from $660 \pm 5$ °C to $1106 \pm 5$°C. At each temperature step two hyperspectral Raman images ($100 \times 100 \ \mu m$, 1 $\mu m$ step size, SWIFT 1.6 $\mu m$/s) were subsequently recorded within a dwell time of four hours.

Above $884 \pm 5$°C, dicalcium silicate $\alpha'_L$-$Ca_2SiO_4$ crystallized by the solid-state reaction between CaO and quartz, which was already observed in experiments with calcite instead of lime (cf. Section 3.2). In both experiments, $\alpha'_L$-$Ca_2SiO_4$ did not only nucleate and grow at visible grain boundaries with quartz grains (Figure 13) but also as many small grains within the CaO matrix. Most likely $SiO_2$ needed for the formation of dicalcium silicate derived from small invisible quartz particles that are more reactive than the larger ones due to their larger surface area. Such a grains size effect was also reported for feldspar by Diella and co-workers, who studying the effect of particle size distribution in Na-feldspar/kaolinite system [75] and is consistent with Oswald ripening that is described by the

Ostwald-Freundlich equation [76] in which surface tension causes small particles to dissolve and larger ones to grow. With increasing time (Section 3.2) or temperature (Section 3.3) dicalcium silicate rims have grown around larger quartz grains (Figures 13 and 16). However, with further increasing temperature (and time) the reaction direction seems to turn over and the dicalcium silicate phase clearly migrated inwards into the quartz grain and partially replaced it (Figure 16). This turn over can only be explained by a change in the mechanism of formation. Up to $929 \pm 5°C$, a tens of micrometer thick $\alpha'_L$-$Ca_2SiO_4$ was formed at the interface between lime and quartz. This rim separated the educts and formed a barrier between lime and quartz. Fierens & Picquet observed by thermal analysis of a $2CaCO_3 + SiO_2$ (mol ratio) mixture that the maximum of the dicalcium silicate formation occurs around 970 °C at virgin surfaces. They claim that the reaction decelerated because the grains were covered with a protective rim of $Ca_2SiO_4$ and therefore the reaction could continue only by diffusion through this rim [77]. This interpretation agrees with our observations. After the $\alpha'_L$-$Ca_2SiO_4$ barrier was formed the reaction mechanism changed. The new formation of $\alpha'_L$-$Ca_2SiO_4$ was possible only by diffusion of the smaller $Ca^{2+}$ ions through the barrier layer, so the quartz grain could be replaced by $\alpha'_L$-$Ca_2SiO_4$.

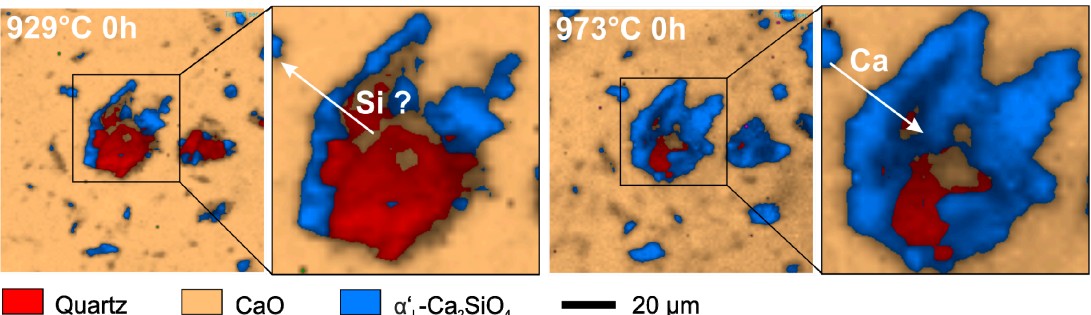

**Figure 16.** Hyperspectral Raman images from a sintering experiment with quartz and lime (CaO), showing grain boundary migration during the solid-state reaction between both reactants. Above $884 \pm 5$ °C the dicalcium silicate $\alpha'_L$-$Ca_2SiO_4$ formed by the direct reaction between lime and quartz as seen in the image taken at $929 \pm 5°C$. At the beginning of the reaction, dicalcium silicate rims seem to have grown around the quartz grains, whereas with increasing temperature (and time) the reaction direction turned over and the dicalcium silicate replaced the quartz grain.

## 4. Future Perspectives

The potential of Raman spectroscopy for *in situ* investigations of sintering reactions is even further expanded by the possibility to image the distribution of $^{18}O$ within and among different phases at the micrometer scale when using $^{18}O$-labeled precursor reactants. For instance, future experiments are planned in which the reactants are labeled with $^{18}O$ in order to find out why the mechanism of the reaction between quartz and lime apparently changed during the experiment (Figure 16). The principle behind analyzing the $^{18}O$ content in condensed matter by vibrational spectroscopy is that the energies or frequencies associated with vibrational motions are dependent on the masses of the vibrating atoms [78]. In the simple harmonic approximation, the vibrational frequency shift is proportional to the square root of the mass ratio between the atoms involved in the vibration [78]. Light stable isotopes, therefore, cause larger isotope effects than heavy isotopes. This mass-dependent frequency of vibrational modes involving the motion of oxygen was already used to semi-quantify the $^{18}O$ content in oxides and silicates such as pyrochlore [79], amorphous silica [80], and feldspar [81], which all formed in $^{18}O$-labeled aqueous solutions as replacement or alteration product [9]. The oxygen isotope $^{18}O$ can potentially also be used as *in situ* tracers to gain insights into mechanistic and dynamic (e.g., transport of matter, crystal growth) aspects of solid state sintering reactions. For instance, using $^{18}O$-enriched calcite or quartz, the involvement of these phases in certain sintering reactions can be recognized by visualizing the distribution of $^{18}O$ among the distinct phases. For this purpose, however, the isotope mass-related frequency shift of a vibrational mode has to be separated from the temperature-related as

well as from a defect-related (chemical) frequency shift in order to correctly estimate the $^{18}O$ content in a mineral at higher temperatures, which adds uncertainty to the quantification of $^{18}O$ [79,81].

## 5. Conclusions

*In situ* hyperspectral Raman imaging provided promising results to *in situ* study solid-solid phase transitions at high-temperatures at a micrometer scale and in real time. This allows the detection of metastable phases and the identification of reverse reactions such as the decomposition of calcite. In addition, the migration of a solid-solid reaction front could be observed *in situ*, which allowed the observation of a turn-over of the reactive direction during the formation of $\alpha'_L$-$Ca_2SiO_4$ from CaO-quartz and thus a change in the reaction mechanism. The same reaction was followed during isothermal sintering at about 850° C for 24 h. A linear increase of $\alpha'_L$-$Ca_2SiO_4$ over time was observed. Accordingly, the fraction of the educts CaO and quartz also decreased linearly. Over time, the average grain size of the newly formed $\alpha'_L$-$Ca_2SiO_4$ grains also increased linearly. A pseudo-zero order behavior of mineral growth was observed in the system, which is most likely due to a CaO excess. The here presented high-temperature Raman imaging approach has a high potential to provide entirely new insights into transport and solid-solid or solid-melt reaction processes during high-temperature sintering and may thus help to monitor and control important technical properties of ceramic materials, including advanced and refractory ceramics, such as densification and grain growth during sintering.

**Author Contributions:** Conceptualization: K.H. and T.G.; Funding acquisition: T.G.; Investigation: K.H., J.K., N.B., and S.Z.; Methodology, K.H., N.B., S.Z., and T.G.; Supervision: T.G.; Visualization: K.H. and J.K.; Writing—original draft: K.H. and T.G.; Writing—review & editing: J.K., N.B., and S.Z.

**Funding:** This research was funded by the deutsche Forschungsgemeinschaft (DFG), grant number GE1094/22-1.

**Acknowledgments:** Many thanks go to T. Schulz and D. Lülsdorf for the technical support with the heating stage.

**Conflicts of Interest:** The authors declare no conflict of interest.

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
