# Peer review of "In Situ Hyperspectral Raman Imaging: A New Method to Investigate Sintering Processes of Ceramic Material at High-temperature"

_applsci, doi:10.3390/app9071310_

Reviewer 1 Report

Manuscript ID: applsci-456271

Title: In Situ Hyperspectral Raman Imaging: A New Method to Investigate Sintering Processes of Ceramic Material at High Temperature

Authors: Kerstin Hauke, Johannes Kehren, Sinje Zimmer, Nadine Böhme, Thorsten Geisler

General comments:

The work reports a novel approach in the study of sintering process at high temperature of ceramic material by in situ hyperspectral Raman imaging. The authors described in a very detailed way the analytical part, the elaboration and correction of data. The focus of the work is interesting and within the aims of the journal. Moreover, the work is clearly presented, well executed and the analytical part is deeply discussed. However, some minor improvements should be added, in particular in the application examples.

Major issues:

1 – I suggest modifying the titles and the organization of the paragraph 2 dividing paragraphs under the titles: materials and method (including also “map programming” and “data reduction, quantification and visualization”) to make easier the organization of the work to the reader.

2 - The authors should improve and clarify the applications of this method linking it to real cases. For example, the application to ancient ceramics could be difficultly appliable if you consider that the surfaces are not flat, the materials are already fired, and that you cannot compare results among ancient ceramics and experimental one with different chemical composition and starting from raw materials completely different.

Other issues:

-Page 15, lines 470-471

Please, correct “newly formed carbonate grains” with “secondary calcite”

Author Response

Response to Reviewer 1 Comments

First of all, we would like to thank both reviewers for their helpful comments which, as we believe, helped to improve our manuscript. 

Major issue 1: I suggest modifying the titles and the organization of the paragraph 2 dividing paragraphs under the titles: materials and method (including also “map programming” and “data reduction, quantification and visualization”) to make easier the organization of the work to the reader.

We changed the title of paragraph 2 from “High Temperature Hyperspectral Raman Imaging" to "Materials und Method". This is in a better agreement with the author’s guideline and makes it easier for the reader to follow the structure of the paper.

However, in our opinion, a subdivision of the paragraph 2 into “Materials” and “Methods” would not be an improvement for the organization of the work to the reader.

Major issue 2: The authors should improve and clarify the applications of this method linking it to real cases. For example, the application to ancient ceramics could be difficultly appliable if you consider that the surfaces are not flat, the materials are already fired, and that you cannot compare results among ancient ceramics and experimental one with different chemical composition and starting from raw materials completely different.

Our method is focused on the study of phase transitions during the firing process of ceramics (and other materials). This method is helpful to basically understand the mechanism of the various solid-state reactions (e.g., the formation of wollastonite or dicalcium silicate), the reaction kinetics, and grain growth with time. These findings are transferable to understand the formation of these phases in nature (during high-T-low-P metamorphoses) or the production of advanced ceramics. For example, the knowledge of the reaction kinetics and grain growth can be useful to improve the firing process of the production of calcium silicate ceramics used as bioactive material in medicine. In summary, the method was not primarily developed to investigate ancient ceramics.

Other issues:

-Page 15, lines 470-471

Please, correct “newly formed carbonate grains” with “secondary calcite”

We replaced “newly formed carbonate grains” by “secondary calcite”.

Reviewer 2 Report

The manuscript used in situ hyperspectral Raman imaging Method to investigate sintering processes of ceramic material at high temperature. It is interesting and innovative. I suggest it can be accepted after minor revision.

The comments are as below:

1.      P.19, line 609: “Most likely SiO2 needed for the formation of dicalcium silicate derived from small invisible quartz particles that are more reactive than the larger ones due to their larger surface area.” I suggest more references should be provided to support the speculation.

2.      P.19, lin3 614-615: “This turn over can only be explained by a change in the mechanism of formation.” I suggest it should be compared with other references about the Ca2SiO4 formation mechanism, such as direct reaction at the interface between SiO2 and CaO and Ca2+ diffusion through the Ca2SiO4 barrier layer.

Author Response

Response to Reviewer 2 Comments

First of all, we would like to thank both reviewers for their helpful comments which, as we believe, helped to improve our manuscript. 

Review 2

Comment 1: P.19, line 609: “Most likely SiO2 needed for the formation of dicalcium silicate derived from small invisible quartz particles that are more reactive than the larger ones due to their larger surface area.” I suggest more references should be provided to support the speculation.

We supported our hypothesis that small quartz grains react faster than large grains with a reference to Diella and co-workers who studied the effect of the grain size of feldspar on mullite formation. Furthermore, this hypothesis is consistent with Oswald ripening, in which surface tension causes small particles to dissolve and larger ones to grow.

Comment 2: P.19, lin3 614-615: “This turn over can only be explained by a change in the mechanism of formation.” I suggest it should be compared with other references about the Ca2SiO4 formation mechanism, such as direct reaction at the interface between SiO2 and CaO and Ca2+ diffusion through the Ca2SiO4 barrier layer.

We thank the reviewer for this helpful suggestion. We now refer to a paper of Fierens and Picquet (1975) whose findings agree with our observations and thus support our interpretation.